# Grit as Perseverance in Physical Activity Participation

**DOI:** 10.3390/ijerph17030807

**Published:** 2020-01-28

**Authors:** Stina Rutberg, Lars Nyberg, Darla Castelli, Anna-Karin Lindqvist

**Affiliations:** 1Department of Health Sciences, Luleå University of Technology, 971 87 Luleå, Sweden; lars.nyberg@ltu.se (L.N.); anna-karin.lindqvist@ltu.se (A.-K.L.); 2Department of Kinesiology and Health Education, The University of Texas at Austin, Austin, TX 78712, USA; dcastelli@utexas.edu

**Keywords:** physical activity, grit, children

## Abstract

Childhood is a critical period for the acquisition of healthy behaviors, and the promotion of sustainable healthy behavior among children is greatly important. Therefore, an increased understanding of the relationship between grit and physical activity in a school context is needed. The purpose of this study is to describe and develop an understanding of students’ and teachers’ awareness and experiences concerning grit as a health-promoting factor. Fifty-five students and three teachers participated in the study. Data were collected through the Short Grit Scale and focus group interviews. There were weak to non-significant correlations between the three teachers’ ratings of their students’ grit and the children’s own ratings. The qualitative results show that children and teachers understood the construct of grit but had slightly different perceptions of it and that grit is not considered to be set in stone. The participants made an association between grit, motivation, meaningfulness, and setting goals. The findings indicate that grit might be an ideal target for making physical activity interventions sustainable.

## 1. Introduction

Childhood is a critical period for the acquisition of healthy behaviors [1], and it is greatly important to promote sustainable healthy behaviors among children [2]. Schools are an important context when ensuring that children adopt health-promoting behaviors, but promoting health might appear to be an added burden when the primary focus of schools is meeting academic standards [3]. Nevertheless, children spend approximately half of their waking hours in school, thus providing an opportunity to promote healthy behaviors for all children, regardless of their life circumstances [1]. 

Participating in regular physical activity entails substantial health benefits for children, even if the improvements in participation are moderate [4]. Unfortunately, a majority of children do not meet current guidelines for physical activity, and the World Health Organization (WHO) states that it is urgent to implement effective programs and investments to intervene in the multiple causes and inequities that perpetuate low physical activity among children [5]. Children´s physical activity can be accumulated in four possible ways: outdoor physical activity, including organized sports or play; physical activity during school hours; physical activity at home; and active transportation to and from places such as school [6]. 

Adding active school transportation such as biking or walking to a child’s daily routine could result in a substantial increase in the number of children meeting the WHO´s health recommendation of 60 minutes of daily physical activity [7]. However, children in Sweden face challenges in their ability to maintain high levels of active school transportation throughout the year because of climatic barriers during winter. Therefore, interventions to increase children’s active transportation throughout the year need to focus on motivational aspects such as self-efficacy and grit [8]. With the long-term goal of achieving a healthy fitness status as a proxy of health, participation in all kinds of physical activity could be viewed as continual work in progress requiring a degree of commitment to achieve one’s desired objective [9]. 

Grit is the adherence and persevering attempts to meet long-term goals. The construct of grit includes perseverance of effort, consistency of interest, goal attainment, and passion [10]. Grit can be divided into two constructs: effort and consistency of interest [11]. Perseverance has a direct and significant effect on one’s intention to be physically active [12], but not on interest. Thus, this study focuses more on the perseverance construct of grit. 

When goal-directed actions reflect one’s values and beliefs, an individual can begin to overcome adversity [13], and grit is a characteristic that can change and develop throughout life along with life experience [14]. Reed, Pritschet, and Cutton [15] have demonstrated that grit is a significant predictor of physical activity, and the construct of grit has been applied to the attainment of physical activity, relative effort or physical activity intensity, and the amount of participation time. In a study on 397 Hispanic female students, grit and school attendance were the primary predictors of school success and may even be as strong of a predictor as physical fitness, which served as a proxy of health [16]. Moreover, grit has also been applied to the attainment of educational goals [17]. In school, educational goals are primarily set by the teacher as an intermediary of the curriculum, but when goal setting is a shared experience and co-regulated through social interactions, one’s pursuit of a long-term goal can be enhanced [18]. 

The construct of grit is not without critics and limitations [19], and further research is warranted to understand the relationship between grit and physical activity in a school context. Given the lack of physical activity and the need to promote children’s health in a sustainable way, children and their teachers were targeted in a Northern Sweden community to examine their perspectives on physical activity. The purpose of this study was to describe and develop an understanding of students’ and teachers’ awareness and their experiences concerning grit as a health-promoting factor.

## 2. Materials and Methods 

### 2.1. Design

This study is part of a series of studies on children and physical activity [8,16,20,21,22]. The current research explores grit-inspired focused groups with regard to students and their engagement in physical activity. Mixed methods were used to explain and complement the knowledge concerning factors that impact students’ physical activity [23]. 

### 2.2. Participants

Information about the project was given to the principals of elementary schools in a municipality in Northern Sweden. Two schools (one class at one of the schools and two classes at the other school) were chosen based on their interest in participating and their previous problems with traffic caused by a large number of parents driving their students to and from school. The authors informed the principals, teachers, students, and parents about the project in person. Sixty-two parents gave informed consent for their students to participate, and all 62 of those students agreed to participate. The study was approved by the Regional Ethical Board in Umeå, Sweden (No. 2018-10-31M). Table 1 provides information about the municipality where the school is located and the participants. 

### 2.3. Procedures

The students and teachers completed surveys, interviews, and focus groups as data sources. The justification for each data source and procedure is described. As part of the study introduction, the teachers had previously been provided with a definition of grit as “perseverance and passion for long-term goals.” The study was approved by the Regional Ethical Board in Umeå, Sweden prior to the start of the research project (Dnr 2018-10-31M).

#### 2.3.1. Quantitative Data Collection Using Grit-S

Both the teachers and students were asked to complete a Swedish translation of the Short Grit Scale (Grit-S), which is a brief self-report scale that measures perseverance and passion for long-term goals [11]. A previous study using confirmatory factor analysis has shown that the Grit–S retains the two-factor structure, consistency of interest, and perseverance of effort of the original grit scale [24]. Furthermore, the two subscales and the whole Grit–S show good internal consistency, it is relatively stable over time, and it does not differ between genders [11]. The scale was translated into Swedish according to principles of good practice for cultural adaptation [25]. A subset of students was rated twice since two of the teachers were involved in the same classes. 

#### 2.3.2. Qualitative Collection Using Focus Groups and Individual Interviews

Focus group interviews were also used as a data source. Using a semi-structured interview protocol, the children were divided into four focus groups of 4 to 5 students each group. The division was performed by their teacher in a manner that would likely facilitate an open and lively discussion. The interview guide included questions to the students concerning their experience with the consistency of interest and the perseverance of effort. To deepen the answers, follow-up questions were asked, such as, “Could you tell me more?” and, “How did you experience that?” [26]. 

Data were also collected from the teachers using individual interviews. We constructed a similar semi-structured interview guide like the one used in the focus groups with questions concerning the teacher’s experience regarding their student’s consistency of interest and perseverance of effort. To deepen the answers, follow-up questions were asked in a similar way as for the focus groups. Two of the authors performed both of the focus groups and the individual interviews, which were carried out in a room without distractions at the school. The sessions lasted between 30 and 45 minutes, and they were sound-recorded and transcribed verbatim.

### 2.4. Data Analysis

The quantitative data collected from Grit-S are presented using the means, standard deviations, and 95% confidence intervals as standard descriptors. The relation between teachers’ and students’ ratings were assessed using linear regression analysis while controlling for sex. The statistical analyses were performed using SPSS version 23 (SPSS Inc., Chicago, IL, USA 2015).

The qualitative data collected from the focus groups and interviews were analyzed through a qualitative content analysis inspired by Graneheim and Lundman [27]. The analysis began with reading the transcribed text several times to obtain a sense of the overall data. The text was then analyzed for meaning units that addressed the aim of the study. The meaning units were coded and sorted into preliminary categories. During this phase, the authors strived to stay close to the text. The preliminary categories were compared and contrasted, and the final results were derived through several discussions between the authors. Quotations were used to strengthen the credibility of the study.

## 3. Results

### 3.1. Quantitative Findings

The students’ mean self-rated grit score was 3.7 with a narrow 95% confidence interval (3.6–3.8) (Table 2). There was a small but statistically insignificant difference between the mean ratings of girls and boys. There were weak to non-significant correlations between the three different teachers’ ratings of their students’ grit and the children’s own ratings (Table 3). The correlations were not influenced by the sex of the children. 

Teachers’ grit estimations showed higher spreads and somewhat lower means than children’s ratings. Correlations were weak to non-existent. Correlations were not influenced by the children’s sex. 

### 3.2. Qualitative Findings

The results of the qualitative analysis formed one main theme and two subthemes. The quotes are labeled with teacher (T) or child (C) and given a participant number, as well as M for male and F for female.

#### 3.2.1. Grit is not Set in Stone

All participants talked about grit as something that could vary from situation to situation and throughout life, thereby not being set in stone. The teachers talked about the students’ grit in comparison to schoolwork, and the children talked about it mostly in connection with sports. Fun seemed to be important for developing long-term interest, but the concept is complex. Fun could be accomplished through social support, having challenging tasks, and having self-efficacy, which supported long-term interest. The effort was benefitted by acknowledging meaningfulness and having goals. Both the teachers and the students recognized that sports could be an environment where it is easier to have clear goals and meaningfulness than in school, but both of these areas have the potential to increase grit. 

##### Having Fun Leads to Long-Term Interest

The students talked about having fun as the most important factor for them in developing long-term interest. All students had leisure activities, they had been performing them for a long time, and most of them had participated since they were very young. 

C1F: My interest is soccer, and I think it is great fun, so that is why I play.C3F: Basketball and singing are fun; I would not want to quit. I have the motivation to continue and be better at it.C2F: Yes, it is fun and you want to learn more.

Doing things together with others and having fun with peers sparked long-term interest among the children. Social support from friends could also motivate them to continue with a sport if they did not feel that they had made any progress. The students mainly talked about long-term interest in sports and other leisure-time activities. 

In contrast, the teachers primarily talked about their students’ grit and long-term interest in regard to school. They perceived that the students had more grit when it came to sports activities than school. They thought that school work is not often associated with fun and is something that has to be done, but a sport is a leisure-time activity that is voluntary and something that is chosen. According to the teachers, it is their responsibility as educators to trigger the students’ grit, their approach can kill interest, or they can inspire the students. They explained that it is more common for students to be interested in a new subject, but when the excitement for the novelty has faded, it is up to the teacher to keep them interested. In the teachers’ experience, they could keep the “spark” alive by varying their teaching methods, and if they let the students participate in the planning and have an influence, there would be an even greater chance that they would show grit and remain interested.

Furthermore, it is important for the assignments to be challenging; if a capable student is given a task that is too easy, their interest sometimes fades. The students agreed, and many of them indicated that they became more interested when they were told that a task would be rather challenging. The teachers highlighted making learning fun, but they emphasized that fun was not the goal; it is a way of maintaining long-term interest. According to the teachers, it is also important to have contrasts during a school day. If fun and play are the only things that are emphasized at school, there is a risk that the fun would wear off and that the students would lose interest anyway. Furthermore, the teachers brought up their own experience that living life is not always fun and easygoing. Having grit during these moments in life is important and something one needs to learn, and the sooner the better.

T1M: We who have lived life know that everything is not fun. You have to stay with it and not be a quitter. 

The teachers also experienced a clear connection between sports activities and school and that there are things children learn when participating in sports. One example is taking responsibility for one’s own development, which might be transferable to the school context and later on in life. Furthermore, the teachers thought that there is an association between grit, self-efficacy, and motivation. A student with low self-efficacy quickly asks for help if they run into adversities and they do not trust their own ability to solve a problem. If students experience that they can overcome difficulties, their self-efficacy increases and they become more motivated. 

T3F: You can practice coping with adversities. How did I solve it the last time, how do you proceed? One must not be afraid to fail. You have to dare to win.

The students agreed with this and expressed that training, concentration, patience, and believing in oneself were key elements. The teachers thought that grit can be influenced by education, but it is not possible to teach grit; it is done by integrating the concept while teaching. One of the teachers described how early in her career, she used to work with rewards to maintain students’ long-term interest, but she does not do that anymore since it did not work in the long run in her experience. The students became so use to receiving rewards that it was “business as usual” and they needed to obtain greater and greater rewards to maintain motivation. The teachers all agreed that acknowledging progress was an efficient way of maintaining the students’ interest and building grit.

#### 3.2.2. Meaningfulness and Goal Setting Regulate Effort

According to the students, they need to be able to imagine the future and understand what their actions and efforts can lead to so that they could see the purpose of it all. This was clear to them in the context of sports, but at school, the students did not clearly see what their efforts would lead to. The teachers thought that using examples from sports might help students to develop awareness and ultimately stronger grit in regard to their school work. They sometimes used sports as an example to explain why the students have to stick to activities in school and repeat something many times. They perceived that the students understood this when they used such an explanation.

T2F: I think of a student who had some problems in math class and is active in a team sport who said, “This was so difficult in the beginning, but then I thought as I do in my sport, I have to train these things.” He has changed his attitude and gone forward a lot in school. His own development is huge. I was impressed.

When the students experienced that something was meaningless, they were not willing to put any effort into the task, and the teachers had the same feeling. Moreover, the teachers perceived that it is their responsibility to convey the meaningfulness of the tasks at school, which is more easily done with more experienced students. When a new concept is being taught, it is difficult to show the meaningfulness compared to when the students have acquired some knowledge. Furthermore, they thought that the students’ ability to execute grit is far more important than their talent, and the teachers’ role is to acknowledge the student’s effort. 

One strategy used by the teachers to support the students when there is a risk of them encountering obstacles is having them work in small groups or pairs—never alone. By doing so, they can help each other to clear obstacles, and there is also great value in being able to explain something to a peer. 

According to the teachers, the students sometimes set goals for their sports achievements, but very few of them set goals for themselves in regard to school-work. The teachers try to convey what the goal is and they think that teachers today are more inclined to be specific in regard to goals, which makes the students more willing to put effort into difficult things. However, there is a difference between various school subjects in the way that this is accomplished; for example, it can be easier to set a goal in mathematics than in the arts. The students experience that goals and competition might trigger their grit and willingness to work hard.

C11F: So I think about how it will be when I have succeeded.C12F: Yes. It can be tough now, but it will be better later.C13F: You get really focused when you set a goal. You really want to do it. Then you become extra focused and try harder.C11F: I did not want to be worse than the boys on one thing. I wanted to do it because I wanted to show that I was as good as them, so I just went on until I succeeded, which did not take a long time.

Both the teachers and the students agreed that grit is something that can be achieved. The teachers thought that the students’ grit increases with age. It was perceived to be harder to maintain the perseverance of effort with younger children since they become frustrated more easily.

T3F: In the fourth grade, oh how they can cry over school-work. But in sixth grade, they have finished crying.

Most students also believed that dealing with setbacks can be handled in several different ways and that the ability to handle them can develop with age. 

C14M: When you are younger, you find something else to do if you fail. You just want to do something else. But when you get older, then you continue until you succeed.C16M: When you finally succeed, you know that it pays off and you are more willing to try something else that is hard.C15M: Yes, nobody is good at something if they have not practiced and exercised.

The teachers thought that their role is to help students to develop their grit and that school and sports activities are good venues to develop grit through health-promotion interventions. However, according to the teachers, students have a harder time coping with setbacks than they did a couple of years ago, and a reason for that is that the parents are more overprotective and act as “lawnmowers.” Benevolent parents want to pave the way for their children, and as a result, the children are not used to putting in the necessary effort themselves. Both teachers and students agree that parents can help their children to be more persistent and keep going even if something requires more effort by asking questions and showing interest without applying too much pressure.

T2F: Parents are there and eliminate the adversity. It is in parenting today; a good parent is like that. Misguided goodwill. Like driving them to school and doing the homework for your child. And then it hurts when you do not have a full score on the exam. What is good parenting today was not good parenting when I started 30 years ago. 

## 4. Discussion

The focus of this study was to describe and develop an understanding of students’ and teachers’ awareness and experiences with grit as a health-promoting factor. The results indicate that teachers and students both understood the construct of grit, although they had slightly different perceptions of it, for instance, the result from the children mirror their whole life situation while the teachers only refer to what they can observe in the school-context. The findings of the qualitative analysis could therefore complement and might explain the weak correlation between teachers’ and students’ results in the quantitative analysis.

The weak correlation contrasts with the conclusions of Duckworth and Quinn [11], who suggest that grit can reliably be assessed by different informants as they found a medium to large correlation between students, family members, and peers. However, the strengths of their correlations (*r* = 0.37 to 0.47) were not more than medium to large, resulting in determination coefficients less than 25% and indicating low reproducibility. Thus, it seems they have overestimated the reliability results of their reported correlation results. Taken together, these findings do not support the assumption that grit ratings by different informants could be used interchangeably. 

Our findings indicate that grit is a characteristic that can change and develop throughout life in line with life experience, which is supported by Hoeschler et al. [14]. Furthermore, our findings show that both teachers and students talked about grit in conjunction with physical activity, and analogies with sports were judged as valuable ways to introduce methods to overcome barriers or adverse life events. Cosgrove et al. [16] conclude that physical activity is one way of developing grit as children learn to set goals and achieve them by overcoming adversities. One study performed in our overarching project examined active school transportation with a focus on winter conditions. The results showed that because of the cold and snow, walking or biking during winter was not easy and required commitment and grit [8]. Furthermore, that study showed that succeeding with this commitment to utilize active school transportation during winter made the students believe that they could accomplish many other things in life by just adopting the correct mindset [8]. 

Both teachers and students believed in enjoyment as a way to promote consistency of interest and that having fun was an important element for long-term engagement. It is a well-known fact that children do not consciously participate in physical activity for health benefits [28]. Instead, they choose to use their leisure time for physical activity because it is an enjoyable way to spend time with their friends [29,30]. The teachers believed that making learning fun was important but not the main goal. According to their experience, life is not always fun and easygoing, and having grit during these moments of life is important and something one needs to learn. This is similar to another study showing that parents perceived the young generation as being driven too much by enjoyment, which might become a threat to their health [31]. 

According to the teachers, letting the students participate in the planning and have influence is an opportunity to maintain their long-term interest and promote grit. The concept of empowerment and including participants are key elements in the sustainability of health promotion [8,32]. Kirk et al. [33] conclude that there is a need to work toward creating schools that empower, engage, and excite students to learn and teachers to teach. This is also something that is emphasized by the WHO, which calls for initiatives that promote the engagement of youth, which will be vital to strengthen the opportunities for physical activity in all communities. They state that such actions will improve the health of this generation and future generations, as well as support the achievement of the 2030 Sustainable Development Goals [5].

Our study shows that there might be a relation between grit, self-efficacy, and motivation. According to the teachers, students’ self-efficacy and motivation increase when they overcome difficulties. In a study on high-school students, Muenks et al. [34] found that the perseverance-of-effort component was more strongly correlated with self-efficacy than the consistency of interests. Perseverance of effort emerged as a significant predictor of end-of-semester grades, but the consistency of interests did not [34]. Usher and colleagues revealed that self-efficacy partially or fully mediated the relationship between grit and school outcomes. They concluded that to improve student performance, teachers should target self-efficacy rather than grit [35]. 

Crede, Tynan, and Harms [19] performed a meta-analysis of grit literature and concluded that grit is only moderately correlated with performance, but it is strongly correlated with conscientiousness. They also found that perseverance of effort had significantly stronger criterion validities than the consistency of interest and that perseverance of effort explains variance in academic performance even after controlling for conscientiousness. They call for a greater focus on the perseverance facet, greater rigor in scale development, and a more nuanced approach in study design to help future grit researchers to develop boundary conditions for grit and its role in performance and success [19].

Our findings show that meaningfulness and being able to imagine the future are vital in relation to the perseverance of effort. Meaningfulness is one of three elements in the sense of coherence [36], which can be influenced through interventions [37,38]. However, there is currently no clear understanding of the mechanisms underlying the sense of coherence [36], and considering grit might be a fruitful avenue to consider when taking on the challenge of determining what is needed in health-promotion activities to strengthen the sense of coherence. Furthermore, given the teacher and student perceptions about goal-directed behavior, the factors of the perseverance of effort may be an ideal target for physical activity intervention.

One limitation of this study is that it was performed with two classes at two relatively small schools located in a small town, which may have influenced the participants’ beliefs and opinions. As compared to data presented regarding adolescents with a mean age of 14 years [11], the mean grit ratings of children in our study were slightly higher and showed less spread. Due to the complex interaction between socioeconomic, environmental, and cultural factors, it is important to consider these factors when investigating concepts such as grit. Future evaluations need to include a wider target group in regard to these factors.

## 5. Conclusions

The findings of our study showed that teachers and students both understood the construct of grit, although they had slightly different perceptions of it. The study also indicated that grit is a characteristic that can change and develop throughout life in line with life experience. Furthermore, our findings showed that both teachers and students talked about grit in conjunction with physical activity and that there might be an association between grit, motivation, meaningfulness, and setting goals. The findings also indicate that it might be fruitful to put a greater focus on the perseverance facet, as meaningfulness is vital in relation to the perseverance of effort. Furthermore, focusing on the perseverance of effort may be an ideal target for making physical activity interventions sustainable. 

## Figures and Tables

**Table 1 ijerph-17-00807-t001:** Characteristics of the context, students, and teachers who participated in the study.

Context	This study was performed within a three-year public health project concerning students’ active school transportation. The study was performed in a municipality with approximately 80,000 inhabitants situated in the northern part of Sweden. The three classes consisted of 64 students (29 boys and 35 girls).
Students participating in the first data collection	55 students (23 boys and 32 girls) aged 11-13 years (mean ± SD: 11.3 ± 0.8 years) from both schools participated in the quantitative data collection.
Students participating in the second data collection	19 (8 boys and 11 girls) out of 22 students from one of the schools participated in the qualitative data collection.
Teachers	Two female teachers and one male teacher with over 15 years of teaching experience participated in both the qualitative and quantitative data collection.

**Table 2 ijerph-17-00807-t002:** Grit scores from students’ own ratings.

Statistics	Girls (*n* = 32)	Boys (*n* = 23)	Total (*n* = 55)
Mean	3.6	3.8	3.7
SD	0.3	0.4	0.4
95% CI	3.5–3.7	3.6–4.0	3.6–3.8

**Table 3 ijerph-17-00807-t003:** Concordance between teachers’ estimation and children’s ratings.

Statistics	Teacher 1 (*n* = 19)	Teacher 2 (*n* = 36)	Teacher 3 (*n* = 36)
Mean ± SD teacher estimation	3.4 ± 1.0	3.1 ± 0.9	3.7 ± 1.0
Mean ± SD corresponding children’s ratings	3.7 ± 0.4	3.7 ± 0.3	3.7 ± 0.3
Correlation	0.53 (*p* = 0.020)	0.25 (*p* = 0.148)	0.03 (*p* = 0.865)

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
