# Peer review of "Grit as Perseverance in Physical Activity Participation"

_ijerph, 2020, doi:10.3390/ijerph17030807_

Round 1

Reviewer 1 Report

This article is valuable because of the search for the importance of grit during promotion of sustainable healthy behavior among children. The article fits in  the development of today's education, in which social competences play a huge role in preparing young people to undertake regular physical activity. Identifying the student's needs in striving to achieve the set goals allows for the urgent implementation of effective programs, such as increasing physical activity of appropriate intensity and duration.

The only note  I have is regarding reference numbers in the text that do not match the reference list. The reason for this is that the first number is assigned the title "References" in the references list.

Congratulations to the authors of the article.

Author Response

This article is valuable because of the search for the importance of grit during promotion of sustainable healthy behavior among children. The article fits in  the development of today's education, in which social competences play a huge role in preparing young people to undertake regular physical activity. Identifying the student's needs in striving to achieve the set goals allows for the urgent implementation of effective programs, such as increasing physical activity of appropriate intensity and duration.

The only note  I have is regarding reference numbers in the text that do not match the reference list. The reason for this is that the first number is assigned the title "References" in the references list.

Congratulations to the authors of the article.

Thank you so much, we also find this article valuable and important as a starting point for future studies. We appreciate the comment about the references, and have corrected the error.

Reviewer 2 Report

Very interesting work, which showed complexity in its execution. The mixed methodology presented is correct and appropriate to the proposed study design. The limitations are evident and presented by the authors, regarding the sample size and social context of the city to which the evaluated children are inserted. Although it is basically a case study, it reflects very well the results obtained considering the theoretical review presented. The triangulation of quantitative and qualitative results could be further explored. We find the FIT indices resulting from the applied factor analysis to be missing. It would also be very interesting to apply Hayes's mediation models, which could have been tested by taking into account the qualitative results obtained and by taking advantage of the quantitative data available to them. The discussion describes the possibility of innumerable base variables that allow GRIT variables to present themselves as mediators or consequential variables in models to be tested. It seems to us that the concept of GRIT can be very interesting to be tested along with motivational determinants, namely Deci and Ryan's theory of self-determination. This link is reflected in the discussion "According to the teachers, letting the students participate in the planning and have an influence is an opportunity to maintain their long-term interest and promote grit." It will also be interesting to test this methodological design in adolescents because the ages of the students evaluated mark a transition in the students' self-concept, and the concept of GRIT may also be influenced or in the process of alteration in this age group. A review of the bibliographic writing standards required by the journal is recommended.

Author Response

Comments and Suggestions for Authors

Very interesting work, which showed complexity in its execution. The mixed methodology presented is correct and appropriate to the proposed study design. The limitations are evident and presented by the authors, regarding the sample size and social context of the city to which the evaluated children are inserted. Although it is basically a case study, it reflects very well the results obtained considering the theoretical review presented. 

We are grateful for your interest and judgement of our study 

The triangulation of quantitative and qualitative results could be further explored. 

We have added text in the discussion part that reflect one more aspect of the triangulation of the quantitative and qualitative result.  We do not think that it is possible to do a make a triangulation in the result part of this study, as we have different data analysis methods for the two parts.

We find the FIT indices resulting from the applied factor analysis to be missing. 

Thank you for this comment; it made us aware that a sentence in the data collection section could be interpreted as we had done such an analysis. We have rewritten this sentence to make this more clear.

 It would also be very interesting to apply Hayes's mediation models, which could have been tested by taking into account the qualitative results obtained and by taking advantage of the quantitative data available to them.

We agree with you that applying Hayes´s mediation model is interesting in future studies. However, we do not understand how this model could be used in this study. As we understand it, it requires big data material.

The discussion describes the possibility of innumerable base variables that allow GRIT variables to present themselves as mediators or consequential variables in models to be tested. It seems to us that the concept of GRIT can be very interesting to be tested along with motivational determinants, namely Deci and Ryan's theory of self-determination. This link is reflected in the discussion "According to the teachers, letting the students participate in the planning and have an influence is an opportunity to maintain their long-term interest and promote grit." It will also be interesting to test this methodological design in adolescents because the ages of the students evaluated mark a transition in the students' self-concept, and the concept of GRIT may also be influenced or in the process of alteration in this age group.

Thank you for these valuable suggestions, we will consider expanding the population to adolescents in future studies.

 A review of the bibliographic writing standards required by the journal is recommended.

We have reviewed the bibliographic and made changes to the errors we found.